# Exogenous Application of Low-Concentration Sugar Enhances Brassinosteroid Signaling for Skotomorphogenesis by Promoting BIN2 Degradation

**DOI:** 10.3390/ijms222413588

**Published:** 2021-12-18

**Authors:** Huachun Sheng, Shuangxi Zhang, Yanping Wei, Shaolin Chen

**Affiliations:** 1Biomass Energy Center for Arid and Semi-Arid Lands, Northwest A&F University, Xianyang 712100, China; joker@nwafu.edu.cn (S.Z.); weiyanping@genecompany.com (Y.W.); 2College of Life Sciences, Northwest A&F University, Xianyang 712100, China

**Keywords:** exogenous sucrose, brassinosteroid (BR), skotomorphogenesis, BZR1 accumulation, BIN2 degradation, TOR

## Abstract

In plants, seedling growth is subtly controlled by multiple environmental factors and endogenous phytohormones. The cross-talk between sugars and brassinosteroid (BR) signaling is known to regulate plant growth; however, the molecular mechanisms that coordinate hormone-dependent growth responses with exogenous sucrose in plants are incompletely understood. Skotomorphogenesis is a plant growth stage with rapid elongation of the hypocotyls. In the present study, we found that low-concentration sugars could improve skotomorphogenesis in a manner dependent on BR biosynthesis and TOR activation. However, accumulation of BZR1 in *bzr1-1D* mutant plants partially rescued the defects of skotomorphogenesis induced by the TOR inhibitor AZD, and these etiolated seedlings displayed a normal phenotype like that of wild-type seedlings in response to both sucrose and non-sucrose treatments, thereby indicating that accumulated BZR1 sustained, at least partially, the sucrose-promoted growth of etiolated seedlings (skotomorphogenesis). Moreover, genetic evidence based on a phenotypic analysis of *bin2-3bil1bil2* triple-mutant and gain-of-function *bin2–1* mutant plant indicated that BIN2 inactivation was conducive to skotomorphogenesis in the dark. Subsequent biochemical and molecular analyses enabled us to confirm that sucrose reduced BIN2 levels via the TOR–S6K2 pathway in etiolated seedlings. Combined with a determination of the cellulose content, our results indicated that sucrose-induced BIN2 degradation led to the accumulation of BZR1 and the enhancement of cellulose synthesis, thereby promoting skotomorphogenesis, and that BIN2 is the converging node that integrates sugar and BR signaling.

## 1. Introduction

In plants, sugars serve not only as primary suppliers of energy and cell wall materials, but also as potent signal molecules involved in growth, development, adaptation, and reproduction [1,2,3]. Plants respond to sugar signals via multiple signaling pathways that are either directly mediated by diverse sensors or indirectly induced by energy and metabolite sensors. Hexokinase1 (HXK1), a glucose-phosphorylating enzyme that serves as an intracellular glucose-sensing protein independent of its enzymatic function, is the first demonstrated example of this type of protein in plants [4,5]. In addition to direct sensing by the HXK1 sensor, intracellular sugar levels can be perceived as metabolic input by energy-sensing regulators, such as sucrose-nonfermentation1-related protein kinase1 (SnRK1) and the target of rapamycin (TOR) kinase. SnRK1 is an evolutionarily conserved energy sensor kinase that orchestrates global gene expression changes in response to sugar/energy deprivation in plants [6,7,8], while TOR is activated in nutrient-rich conditions such as after exogenous addition of sucrose or glucose [9,10,11]. Moreover, it was proposed that sugar-activated TOR phosphorylates its substrates (such as S6Ks) to initiate the transmission of sugar signals [2,10].

Sucrose, a major product of photosynthesis, is the main form of sugar that is systemically transported from source to sink, and most plants have a range of sucrose transporters (SUTs) that play different roles in the loading and unloading of sucrose in diverse tissues. However, although the exogenous application of sucrose results in reductions in the mRNA levels and protein activities of SUTs [12], the hypothesis proposing SUTs as sucrose sensors remains contentious. Indeed, identifying sucrose sensors can present a considerable challenge, given that sucrose metabolites such as glucose, fructose, and trehalose 6-phosphate (T6P) can similarly trigger sugar signal transduction [1,13,14]. Nevertheless, despite the current lack of success in identifying sucrose sensors, a compelling amount of diverse information has accumulated to indicate that sucrose signaling regulates plant development and metabolism with respect to flowering [15], nitrogen metabolism [16], anthocyanin accumulation [17], and photosynthesis [18].

It is well established that the phytohormone brassinosteroid (BR) plays particularly important roles in plant growth and development, including normal plant cell expansion and elongation. The BR receptor BRASSINOSTEROID-INSENSITIVE1 (BRI1), a leucine-rich repeat receptor-like protein kinase (LRR-RLK) localized in the plasma membrane, has been shown to be necessary for the initiation of BR signaling. BR-bound BRI1 or activated BRI1 heterodimerizes with BRI1-ASSOCIATED KINASE1 (BAK1) to reduce the activity of BRASSINOSTEROID-INSENSITIVE2 (BIN2) [19]. In the absence of BR, BIN2 phosphorylates and inactivates two transcription factors, BRASSINAZOLE-RESISTANT1 (BZR1) and BRI1-ETHYL METHANESULFONATE-SUPPRESSOR1 (BES1, also referred to as BZR2), thereby inhibiting their DNA-binding capacity and ultimately leading to their cytoplasmic retention or degradation [20]. Conversely, in the presence of BR, PROTEIN PHOSPHATASE 2A (PP2A)-mediated dephosphorylation rescues the activity of BZR1 and BES1 [21,22], thereby activating BR-induced genes expression and promoting cell elongation. 

The authors of previous studies have proposed the interaction between BR and sugars. In *Arabidopsis*, for example, the BR biosynthesis-defective mutant *dwarf1-6*, as well as a *CPD*-antisense line, showed a clear reduction in starch content and assimilatory capacity [23]. Moreover, the expression of the *CPD* gene was repressed by treatment with high levels of exogenous sugars [24], whereas in tomato plants, the sucrose transporter SISUT2 directly interacted with BAK1 and the BR signaling inhibitor MSBP1 to regulate arbuscular mycorrhiza formation [25]. In addition, SlSUT2 RNAi plants exhibited a dwarfed phenotype, with flowers characterized by male sterility, and these characteristic phenotypes could be partially rescued by BR treatment [26].

In the dark, seedling establishment following germination is called skotomorphogenesis. This process is characterized by a small unopened etiolated cotyledon, rapid elongation of the hypocotyl, and retarded primary root growth [27]. BR is a key hormone responsible for skotomorphogenesis in plants. In the present study, we focused on the proposed cross-talk between sucrose and BR signaling with respect to skotomorphogenesis in the dark and established that BIN2 serves as a converging node that integrates the cross-talk between sugar and BR signaling in the regulation of plant growth. Furthermore, we found that sucrose reduced BIN2 levels, thereby increasing the accumulation of dephosphorylated BZR1 and enhancing cellulose synthesis. These findings accordingly provide evidence that sugar promotes BR signaling, which in turn contributes to the enhanced growth of etiolated seedlings in the dark.

## 2. Results

### 2.1. BR Biosynthesis Is Required for Sucrose Promotion of Skotomorphogenesis

Under dark growth conditions, germinated seeds undergo skotomorphogenesis, with rapid elongation of the hypocotyls. Hypocotyl growth is dependent on a sufficient supply of sucrose, which is required as a source of building blocks for cell wall construction and signal molecules to induce hypocotyl elongation. Exogenous application of low-concentration sucrose or 24-epibrassinolide (eBL) can promote a marked increase in hypocotyl elongation of etiolated seedling, whereas high-concentration supplementation of either sucrose or eBL has the effect of inhibiting skotomorphogenesis (Figure 1A,B). Having confirmed that the inhibitory action of high-concentration sucrose is not attributable to an osmotic effect (Figure 1C), we found that low-concentration sucrose exacerbated the suppressive effect mediated by high-concentration eBL (Figure 1D), rather than alleviating it. Given that exogenous sucrose is known to have effects on etiolated seedling growth like those observed in response to the application of exogenous eBL, it is plausible that there exists a cross-talk between the signaling pathways of sucrose and BR. In this regard, we focused only on the promotion of skotomorphogenesis by low-concentration sucrose (30 mM). To further determine the relationships between BR and sugar signaling, we treated the BR biosynthetic-defective mutant *de-etiolated 2* (*det2-1*) with sucrose and/or eBL. The results showed that the skotomorphogenesis promotion effect of sugar was blocked when endogenous BR was un-guaranteed (Figure 1E,F and Appendix A). However, when supplied with exogenous eBL, the sucrose sensitivity of *det2-1* mutant seedlings was partially rescued (Figure 1E,F). These observations thus tend to indicate that BR biosynthesis is required for the promotion of skotomorphogenesis by sucrose, which is consistent with the findings of sucrose-promoted hypocotyl elongation of de-etiolated seedlings under dark growth conditions [28]. 

### 2.2. The Roles of TOR Kinase and BZR1 Protein in Skotomorphogenesis

It has been reported that sugar–TOR signaling promotes an acceleration in the biosynthesis and accumulation of BZR1 to induce hypocotyl elongation of de-etiolated seedlings under dark growth conditions [28,29]. In the present study, we found that exposure of etiolated seedlings to the TOR inhibitor AZD clearly had a suppressive effect on growth and that the effects of sucrose were abolished (Figure 2A). In addition, we observed that BZR1-accumulating *bzr1-1D* mutants displayed lower sensitivity to AZD than the WT specimens (Figure 2B), regardless of sucrose supplementation. To confirm the effects of sugar–TOR signaling on BZR1 accumulation, we examined the levels of *BZR1* gene transcription and performed immunoblot analysis of BZR1 protein in etiolated seedlings using an anti-BZR1 antibody. It was accordingly established that at both the mRNA and the protein levels, the concentrations of BZR1 were considerably higher in sucrose-supplemented etiolated seedlings than in sucrose-deprived etiolated seedlings (Figure 2C,D). Furthermore, the gain-of-function *bzr1-1D* mutant was found to have higher levels of BZR1 compared with the WT seedlings (Col-0) under both sucrose-supplemented and -limited conditions (Figure 2D). However, a phenotypic analysis of *bzr1-1D* and Col-0 etiolated seedlings revealed no significant differences between BZR1-accumulating and WT plants with respect to hypocotyl elongation (Figure 2E), regardless of whether these plants were grown in the presence or absence of exogenous sucrose.

### 2.3. BIN2 and Its Homologs Negatively Regulate Etiolated Seedling Growth (Skotomorphogenesis)

Within the BR signaling pathway, BIN2 lies upstream of BZR1, and its activation causes either the cytoplasmic retention or the degradation of BZR1. In common with BIN2 kinase, its two closest homologs, BIN2-Like1 (BIL1) and BIN2-Like2 (BIL2), are members of the glycogen synthase kinase 3 (GSK3) family [30], and amino acid sequence alignment revealed that these two highly similar BILs show extensive sequence identity with BIN2, despite the presence of additional N-terminal amino acids and different C-terminal tails (Figure 3A). Notably, BIL1 and BIL2 display BES1 phosphorylation activity like that of BIN2 kinase [30]. These observations accordingly tend to reinforce the belief that these three GSK3s are involved in BR signaling. Moreover, given the assumed cross-talk between sucrose and BR signaling, it is plausible that GSK3s participate in the sucrose-induced response.

Interestingly, in the absence of exogenous sucrose, the etiolated seedlings of the *bin2-3bil1bil2* triple-mutant showed an elongated hypocotyl phenotype (Figure 3B), whereas when grown on sucrose-supplemented medium, the triple-mutant was characterized by slightly shorter hypocotyls (Figure 3B), a phenotype similar to that of WT etiolated seedlings treated with both BR and sucrose (Figure 1D). Moreover, the homozygous (−/−) and heterozygous (+/−) gain-of-function *bin2-1* mutants, in which BIN2 is continuously activated, had a shorter hypocotyl and were insensitive to sucrose (Figure 3C, Appendix A). These observations thus provide evidence that BIN2 and its homologs play a negative role in sugar-induced skotomorphogenesis.

### 2.4. Sucrose–TOR Signaling Reduces the Abundance of BIN2

In the absence of BR, BIN2 is activated following protein autophosphorylation, whereas it is degraded in the presence of BR. Thus, the activity of BIN2 and its homologs is inevitably dependent on their protein abundance and phosphorylation status. To examine the effects of sucrose–TOR signaling on the activity of GSK3s in etiolated seedlings, we assessed the mRNA levels of *GSK3*s and the abundance of BIN2 protein based on qRT-PCR and western blot analyses, respectively. Our observations indicated that, whereas sucrose did not promote a reduction in the mRNA levels of BIN2 (Figure 4A), BIN2 protein abundance was reduced in the presence of exogenous sucrose (Figure 4E). Although the expression of both *BIL1* and *BIL2* was induced by sugar (Figure 4B,C), the mRNA levels of *BIN2* were found to be considerably higher than those of the two homologs (Figure 4D), thereby indicating that BIN2, rather than BIL1 or BIL2, plays a major role in sucrose-mediated hypocotyl elongation in etiolated seedlings, which is also consistent with the observed hypocotyl lengths in the gain-of-function *bin2-1* mutant. Moreover, the expression patterns of the three *GSK3* genes and BIN2 abundance were found to be upregulated by the TOR inhibitor AZD (Figure 4A–C,E). Collectively, these results indicate that sucrose–TOR signaling promotes the degradation of BIN2, thereby facilitating skotomorphogenesis.

### 2.5. S6K2 Downstream of TOR Interacts with and Phosphorylates BIN2 to Induce Skotomorphogenesis

Plant BIN2 protein shows a high sequence similarity to human GSK3β, which is phosphorylated by S6K in animals and humans. It has also been reported that S6K2 can interact with and phosphorylate BIN2 in *Arabidopsis* [31]. In the present study, we sought to further confirm whether AtS6K2 interacts directly with AtBIN2 in the sucrose–TOR pathway. To this end, we performed a split-luciferase complementation assay, the results of which clearly indicated that S6K2 interacts with BIN2 (Figure 5A). A previous in vitro kinase assay and mass spectrometric analysis identified three S6K2-catalyzed phosphorylation sites on BIN2 [31], among which S203 was found to be phosphorylated in vivo by proteomic analyses [32] (http://epsd.biocuckoo.cn/ accessed on 21 May 2021) (Figure 5B). Genetic evidence indicated that S6K inhibitor-induced inactivation of S6K2 resulted in a reduction in hypocotyl length and sucrose sensitivity of etiolated seedlings (Figure 5D), thereby indicating that S6K2 functions as a positive regulator of the sucrose-mediated promotion of skotomorphogenesis. Moreover, BIN2 acts as a downstream effector of AtS6K2 to regulate hypocotyl elongation of etiolated seedlings, and a deficiency in GSK3 function could alleviate the growth inhibition induced by the inhibition of S6K (Figure 5E and Appendix A). These observations thus tend to indicate that BIN2 serves as a converging node that integrates the crosstalk between BR and sucrose–TOR–S6K2 signaling in the regulation of skotomorphogenesis.

### 2.6. Effects of Sucrose on Crystalline Cellulose Biosynthesis

The cell wall, the basic skeleton of which is composed of cellulose microfibrils, plays essential functional roles in supporting plant growth [33], and consequently, hypocotyl elongation is assumed to involve cell wall synthesis and remodeling, particularly the biosynthesis of crystalline cellulose. Our examination of the crystalline cellulose contents of etiolated seedlings revealed that sucrose enhanced the crystalline cellulose content of both WT (Col-0 and Ws-2) and *bzr1-1D* etiolated seedlings (Figure 6). Moreover, we detected no significant differences between *bzr1-1D* and Col-0 etiolated seedlings with respect to crystalline cellulose content, regardless of sucrose supplementation (Figure 6A). Most noteworthy in this regard was our observation that the *bin2-3bil1bil2* triple-mutant grown on sucrose-deprived medium was characterized by a clear increase in crystalline cellulose content compared with Ws-2 WT seedlings, with only a weak increase in crystalline cellulose content in the etiolated seedlings grown on medium supplemented with sucrose (Figure 6B). These observations support our contention that sucrose-induced BIN2 inactivation results in hypocotyl elongation of etiolated seedlings and that a synergistic interaction between sucrose and BR contributes to the inhibition of skotomorphogenesis.

## 3. Discussion

When seedlings grown in light are transferred to darkness or seeds germinate in the dark, exogenously applied sucrose significantly promotes hypocotyl elongation. This effect is involved in gibberellin (GA) and BR biosynthesis [28,29,34]. Moreover, it has been reported that the endogenous GA and BR levels are higher in dark-grown seedlings than that in light-grown seedlings [29,35,36]. GA is required for sucrose-induced hypocotyl elongation, but application of GA_3_ was not able to rescue the defects observed in the quadruple mutant pif1pif3pif4pif5, indicating that it is an important molecular pathway for the GA regulation of hypocotyl elongation via the DELLA-medicated regulation of the protein levels of phytochrome interacting factors (PIFs) [37]. Likewise, BR has similar effects on the sugar-mediated promotion of hypocotyl elongation, and accumulation of BZR1 plays a key role in sugar-induced seedling growth in the dark [28,29]. The interaction between PIF4 and BZR1 has been proven [38], suggesting a cross-talk between GA and BR signaling with a role in hypocotyl elongation. Indeed, the interaction between GA and BR in the control of *Arabidopsis* hypocotyl elongation has been reported [35,39]. Herein, we will discuss the cross-talk between sugar and BR signaling during the stage of skotomorphogenesis.

### 3.1. BZR1 Partially Functions in Sugar-Mediated Skotomorphogenesis

BRs govern multiple processes associated with morphogenetic change. In this regard, it was proposed that BR is required for the sugar-mediated promotion of hypocotyl elongation of de-etiolated seedlings under dark growth conditions [28], which is consistent with the findings of the present study (Figure 1). Zhang et al. (2015) also reported that sugar positively regulates BZR1 at both the transcriptional and the translational levels, thereby promoting the growth of de-etiolated seedlings [28]. In a subsequent study, Zhang et al. (2016) presented evidence indicating that the sugar-induced hypocotyl elongation of de-etiolated seedlings is dependent on sugar–TOR signaling under dark growth conditions, which stabilizes BZR1 via the inhibition of autophagy and thus promotes growth [29]. However, in the gain-of-function mutant *bzr1-1D* the hypocotyl phenotype of the *tor* mutant for sugar-induced elongation in the dark was partially restored [29]. Taken together, these observations and our results indicate that the effects of BZR1-mediated transcriptional regulation on skotomorphogenesis are conditional and that BZR1 probably plays only a partial role in sugar-mediated etiolated seedling growth.

### 3.2. Sucrose-Mediated BIN2 Degradation Promotes Skotomorphogenesis

The detected levels of BZR1 or BES1 are considered to serve as indicators of BIN2 activity [40], as a reduction in BIN2 activity promotes the accumulation of both dephosphorylated BZR1 and BES1. It has been proposed that BIL1 and BIL2 play similar roles as BIN2 kinases [30]. Our observations led us to propose that sucrose-mediated GSK3s inactivation may promote skotomorphogenesis and that BIN2, rather than BIL1 or BIL2, plays a major role in sucrose-mediated skotomorphogenesis. This assumption is based on the following reasoning. Firstly, the transcriptional levels of the *BIN2* gene were found to be considerably higher than those of the *BIL1* and *BIL2* genes. Secondly, it has been reported that almost all transgenic plants overexpressing *gbin2-1(E263K)* exhibit a prominent *bin2-1*-like phenotype, whereas only approximately one-third of *gbil1(E295K)* or *gbil2(E293K)* transgenic plants are morphologically similar to the *bin2-1* mutant [30]. Thirdly, the homozygous *bin2-1* mutant is completely insensitive to sucrose.

### 3.3. Putative Mechanisms of Sucrose-Induced BIN2 Inactivation in Darkness

In the presence of BR, the activated BRI1/BAK1 complex phosphorylates and activates BSK (BR-signaling kinase), thereby promoting the interaction between BSK and BSU1 (bri1 suppressor 1). Subsequently, BSU1, switched on via an interaction with BSK, dephosphorylates and inactivates BIN2. In addition, BIN2 activity has been shown to be regulated by modification of specific cysteine, lysine, or serine residues [31,40,41]. Thus, the mechanisms associated with sucrose-induced BIN2 inactivation under dark growth conditions may plausibly involve BR biosynthesis and/or the post-translational modification of BIN2 (Figure 7). Recently, the findings of several studies have indicated that sugar has the effect of increasing BR accumulation in the dark [29] and that low-concentration glucose can enhance the interaction between BRI1 and BAK1 in a manner dependent on BR biosynthesis [42], which leads to the dephosphorylation and degradation of BIN2. It has also been proposed that TOR signaling can activate S6Ks via modified phosphorylation and that activated S6K2 further phosphorylates BIN2 at the S187 and S203 sites to reduce BIN2 activity, thereby regulating the photoautotrophic growth of *Arabidopsis* [31]. Notably, however, S6K1, the closest homolog of S6K2 (Appendix A) does not interact with BIN2 [31], and thus whether S6K1 plays a role in sucrose-promoted hypocotyl elongation of etiolated seedlings warrants further study. In addition to the TOR–S6K2 pathway, sugars can influence BR signaling via histone deacetylase HDA6-induced deacetylation of BIN2 at the K189 site [40]. In this study, however, we did not provide evidence of the involvement of the HDA6-mediated pathway.

Interestingly, exogenous provision of sucrose also increases the accumulation of BZR1 (primarily, phospho-BZR1) in illuminated seedlings (Figure 2B). Other studies have demonstrated that in plants grown in the light, sugar promotes the accumulation of BIN2 protein and reduces BR-responsive growth, via a yet to be identified sugar signaling pathway that functions independently of the sugar sensors HXK1 and TOR [43]. As described above, however, TOR kinase is required for the sugar-mediated promotion of skotomorphogenesis, and its activation by sucrose reduces BIN2 levels in etiolated seedlings. Collectively, these results indicate that the effects of sugar on BR-mediated plant growth differ under dark and light conditions. Similar differences are also observed in root development: under light condition, glucose–TOR signaling reprograms the transcriptome and activates root meristems to elongate root [44], whereas sugar promotion of root elongation is independent of the TOR and BR signaling in the dark (Appendix A).

### 3.4. Sucrose Enhances Cellulose Synthesis for Skotomorphogenesis

Changes in plant morphology, including hypocotyl elongation, are governed to a large extent by the plant cell wall, as turgor-driven plant cell growth is dependent on cell wall structure and mechanics [45]. The cell wall of plants is mainly composed of cellulose, hemicellulose, and pectin, and recently, close links between cellulose synthesis and phytohormone-controlled plant growth have been proposed [46]. In elongating cotton fiber, in which cellulose production is intimately linked to fiber elongation, exogenous eBL was shown to enhance cellulose production by promoting fiber elongation; in addition, exposing plants to the BR biosynthesis inhibitor BRZ was found to cause shortening of these fibers in *Gossypium herbaceum* [47]. Furthermore, in *Arabidopsis*, BR was found to regulate cellulose biosynthesis by controlling the expression of *CESA* genes [48]. Considering the effects of sugar on BR biosynthesis, sugar-induced hypocotyl elongation of etiolated seedlings may be dependent of BR-mediated cellulose synthesis. It has also been reported that sugar and BR can simultaneously regulate the transcript levels of several genes in etiolated seedlings, including cell wall-related genes such as *expansin A17* (*EXPA17*), *expansin-like A2* (*EXPLA2*), *xyloglucan endotransglucosylase6 (XTH6)*, and *cellulose synthase5* (*CESA5*) [24]. Cell wall-related gene expression may be in connection with BES1 transcription factor rather than BZR1 [48,49]. Moreover, in the absence of BR, BIN2 can directly phosphorylate cellulose synthase 1 (CESA1) at the T157 site, and mutation of an A base at this position promotes the growth of etiolated seedlings by enhancing the activity of CESA complexes, thereby indicating that BIN2 functions as a negative regulator of cellulose synthesis and hypocotyl elongation [50]. Based on these observations, we examined the crystalline cellulose content of BR-related mutants and accordingly obtained evidence indicating that the cross-talk between sucrose and BR signaling contributes to the control of cellulose synthesis and in turn to the regulation of hypocotyl elongation of etiolated seedlings. Thus, it is conceivable that sucrose signaling enhances the activity of CESA complexes for cellulose synthesis and skotomorphogenesis [51] via the inhibition of BIN2 activity and that BIN2 degradation can lead to positive effects on hypocotyl elongation through a BZR1-independent pathway (such as BES1- and CESAs-related pathways).

## 4. Materials and Methods

### 4.1. Plant Material and Growth Conditions

For the purposes of this study, we used the Columbia (Col-0) and Ws-2 ecotypes of *Arabidopsis thaliana* as wild-type (WT) controls for hypocotyl and crystalline cellulose measurements. Col-0 was also used as a control in protein and mRNA analyses. The mutants *det2-1* [52] and *bzr1-1D* [53], which are in a Col-0 background, were obtained from the Arabidopsis Biological Resource Center (ABRC, Ohio State University, Columbus, OH, USA), whereas the *bin2-3bil1bil2* triple mutant in a Ws-2 background and the gain-of-function *bin2-1* mutant were kindly provided by Dr Xuelu Wang (Henan University, Kaifeng, China). Surface-sterilized seeds were sown on half-strength Murashige and Skoog (MS) medium supplemented with 0.8% phytagel. Plates were incubated for 3 days at 4 °C to break dormancy and were thereafter placed in a growth chamber in the dark at 22 ± 1 °C, unless stated otherwise.

### 4.2. Hypocotyl Growth Measurements

For hypocotyl growth assays, sterilized seeds were grown on 1/2 MS medium with or without 30 mM sucrose in the dark for 5 days. To assess potential interactions between sucrose and BR, the medium also contained BR (Epibrassinolide) (Sigma-Aldrich, St. Louis, MO, USA), the TOR inhibitor AZD8055 (1 μM) (M1666, AbMole, Houston, TX, USA), the S6K inhibitor LY2584702 tosylate (AbMole, M4864), or a mock solution of dimethyl sulfoxide (DMSO). At the end of the designated incubation period, the etiolated seedlings were carefully removed from the agar plates and placed on a flat surface for photography with a digital camera. The hypocotyl lengths of individual seedlings were measured using ImageJ software (http://rsb.info.nih.gov/ij/, accessed on 10 May 2021).

### 4.3. Protein Extraction and Western Blot Assays

For protein extraction, liquid nitrogen-frozen plants were ground to a powder. Weighed aliquots were mixed with 2× sodium dodecyl sulfate (SDS) buffer [0.125 mM Tris-HCl (pH 6.8), 4% SDS, 20% glycerol, and 2% β-mercapto-ethanol] in a 1:1 *w*/*v* ratio (i.e., 1 mg of tissue powder per mL of buffer). Samples were heated for 10 min at 65 °C and centrifuged at 10,000× *g* for 10 min at 4 °C. The extracted proteins were separated by 10% SDS-polyacrylamide gel electrophoresis, and thereafter, the separated proteins were transferred to PVDF membranes (Millipore, Billerica, MA, USA) using a Trans-blot Turbo blotting system (Bio-Rad Laboratories, Hercules, CA, USA). Prior to probing with the requisite primary antibodies, the membranes were blocked for 1 h at room temperature in blotting buffer (0.5% skim milk in TBST solution), followed by three washes with TBST buffer. For the immunodetection of BZR1 and BIN2, we used anti-BZR1 (R3528-1, Abiocode, Agoura Hills, CA, USA) and anti-BIN2 (AS163203, Agrisera, Vännäs, Sweden) primary antibodies, respectively, at 1:1000 dilutions, which were incubated with the membranes for 1 h at room temperature. For the immunodetection of beta-actin, used as a control, we used a primary antibody (CW0264, CWBIO, Beijing, China) following the same procedure as that described for BZR1 and BIN2.

### 4.4. RNA Isolation and Quantitative Real-Time PCR Analysis

Total RNA was extracted from 5-day-old etiolated seedlings using a RNeasy Plant Mini Kit (QIAGEN, Hilden, Germany). Complementary DNA (cDNA) was synthesized Invitrogen, Carlsbad, CA, USA) and subsequently used as a template for quantitative real-time PCR (qRT-PCR), performed using a CFX96 Real-Time System (Bio-Rad Laboratories, Hercules, CA, USA) in conjunction with a SYBR Premix Ex Taq II Kit (Takara Bio, Otsu, Japan), according to the manufacturer’s procedures. The *Arabidopsis UBQ10* gene was used as an internal control for normalization of the expression levels of the target genes. The sequences of the primers used for amplification in qRT-PCR analyses are listed in Appendix A.

### 4.5. Split-Luciferase Complementation Assay

Liquid LB-grown *Agrobacterium* strain GV3101 harboring the corresponding constructs was harvested by centrifugation. The pellets thus obtained were resuspended in injection buffer (10 mM MgCl_2_, 10 mM MES, pH 5.6, and 100 mM acetosyringone) to a final concentration corresponding to an OD_600_ of 0.5. After incubation at room temperature for 4 h, co-infiltration was carried out using equal volumes of the two different *Agrobacterium* strains carrying the nLUC and cLUC constructs, which were mixed and infiltrated into the leaves of *Nicotiana benthamiana* plants. Two days later, the infiltrated leaves were sprayed with 1 mM d-luciferin potassium salt solution, and the resulting fluorescence was detected using a charge-coupled device (CCD) camera (Princeton Instruments, Trenton, NJ, USA).

### 4.6. Measurements of Crystalline Cellulose Content

The crystalline cellulose content was determined using the method described by Updegraff [54], with minor modifications. Briefly, 5-day-old etiolated seedlings were harvested and then ground to a powder in liquid N_2_. The homogenate was vacuum-dried overnight. The cell wall powder was washed in turn with 70% ethanol, 1:1 (*v*/*v*) chloroform/methanol, and acetone, each time with centrifugation (10,000 rpm, 10 min). Samples (2 mg) of the washed material were resuspended in 400 μL of 2 M trifluoroacetic acid (TFA) solution and then incubated at 121 °C for 90 min. The TFA-insoluble material was separated and dried and then suspended in 1 mL of Updegraff reagent. After incubation at 100 °C for 30 min in boiling water, the samples were centrifuged at 2500× *g* for 5 min, and the remaining crystalline cellulose was dissolved in 72% (*v*/*v*) sulfuric acid. The crystalline cellulose thus obtained was quantified colorimetrically at 625 nm in a spectrophotometer using anthrone reagent [54].

## 5. Conclusions

The findings of this study reinforce the concept that BR biosynthesis and TOR kinase are required for the sucrose-mediated promotion of skotomorphogenesis in the dark. The significance of this study is that we propose a plausible mechanism whereby sugar–TOR–S6K signaling inhibits the activity of BIN2 by promoting its degradation. Taken together with the findings of previous studies, our current observations have enabled us to establish that the degradation of BIN2 leads to the accumulation of BZR1 for transcriptional regulation and enhancement of CESA complex activity for cellulose synthesis, which may synergistically facilitate the promotion of etiolated seedling growth. We accordingly believe that our observations will contribute to gaining a better understanding of the mechanisms underlying sucrose-induced skotomorphogenesis in the dark and promote practical applications of BR and sugar during the stage of skotomorphogenesis.

## Figures and Tables

**Figure 1 ijms-22-13588-f001:**
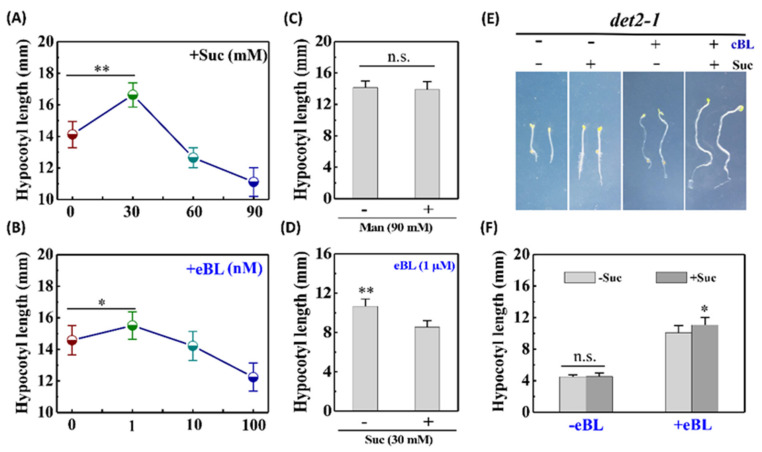
Sucrose promotion of skotomorphogenesis depends on brassinosteroid (BR) biosynthesis. (**A**,**B**) Hypocotyl length of wild-type (WT) etiolated seedlings grown on medium supplemented with different concentrations of sucrose and eBL, respectively. (**C**) Hypocotyl length of WT etiolated seedlings grown on medium with or without 90 mM mannitol. (**D**) Synergistic effects of exogenous sucrose and eBL. (**E**) Images of 5-day-old dark-grown etiolated seedlings of *det2-1* treated with or without 30 mM sucrose ± 1 μM eBL. (**F**) Hypocotyl length of etiolated seedlings measured in those shown in (**E**). Error bars represent standard deviations (*n* > 20). Data were statistically evaluated using Student’s *t*-test, and the significance of differences (**, *p* < 0.01; *, *p* < 0.05; n.s., not significant) is indicated.

**Figure 2 ijms-22-13588-f002:**
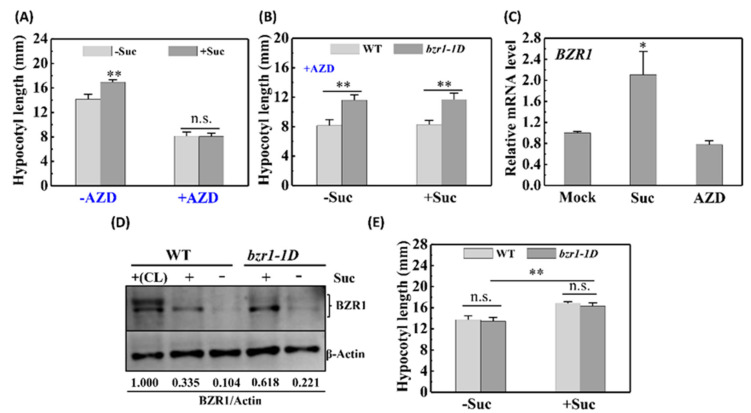
Activation of TOR kinase and accumulation of BZR1 play a role in sugar-induced skotomorphogenesis. (**A**) TOR kinase is essential for skotomorphogenesis. (**B**) Hypocotyl length of wild-type (WT) and *bzr1-1D* etiolated seedlings treated with 1 μM AZD ± 30 mM sucrose. (**C**) Expression pattern of the *bzr1* gene in WT etiolated seedlings treated with 30 mM sucrose or 1 μM AZD. Values are presented as the means ± SD of three independent sets of experiments. (**D**) Immunoblot analysis of BZR1 protein in WT and *bzr1-1D* seedlings grown in the dark on medium with (+) or without (−) 30 mM sucrose. The upper and lower bands indicate phosphorylated and dephosphorylated BZR1, respectively. (CL) indicates that plants were grown in continuous white light. (**E**) Hypocotyl length of WT and *bzr1-1D* etiolated seedlings treated with or without 30 mM sucrose. Error bars for standard deviations (*n* > 20). Data were statistically evaluated using Student’s *t*-test, and the significance of differences (**, *p* < 0.01; *, *p* < 0.05; n.s., not significant) is indicated.

**Figure 3 ijms-22-13588-f003:**
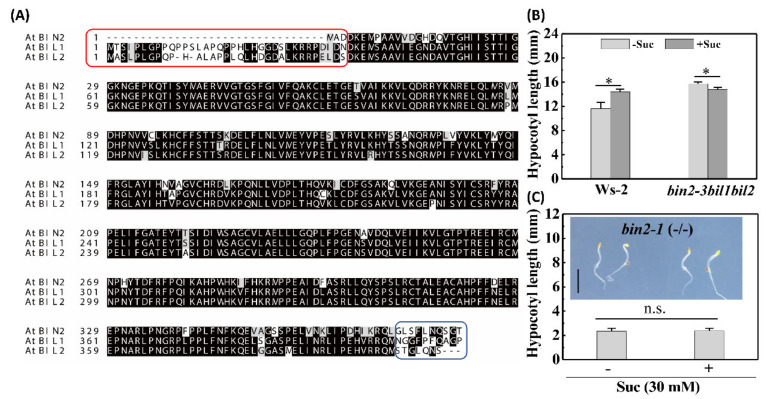
GSK3s function redundantly in etiolated seedling growth. (**A**) Amino acid sequence alignment of BIN2, BIL1, and BIL2 performed using ClustalW2 (https://www.ebi.ac.uk/Tools/msa/clustalo/ (accessed on 21 May 2021)) and displayed using Expasy Boxshade (https://embnet.vital-it.ch/software/BOX_form.html (accessed on 21 May 2021). Highly conserved amino acids are indicated by black shading, whereas gray indicates less highly conserved residues. Additional N-terminal amino acids and different C-terminal tails are boxed in red and blue, respectively. Hypocotyl length of *bin2-3bil1bil2* (**B**) and *bin2-1* (**C**) mutants treated with or without 30 mM sucrose. Error bars for both (**B**) and (**C**) represent the standard deviations (*n* > 18). Inset in (**C**) is the image of 5-day-old dark-grown seedlings of the homozygous *bin2-1*(-/-) mutant. Scale bars represent 3 mm. Data were statistically evaluated using Student’s *t*-test, and the significance of differences (*, *p* < 0.05; n.s., not significant) is indicated.

**Figure 4 ijms-22-13588-f004:**
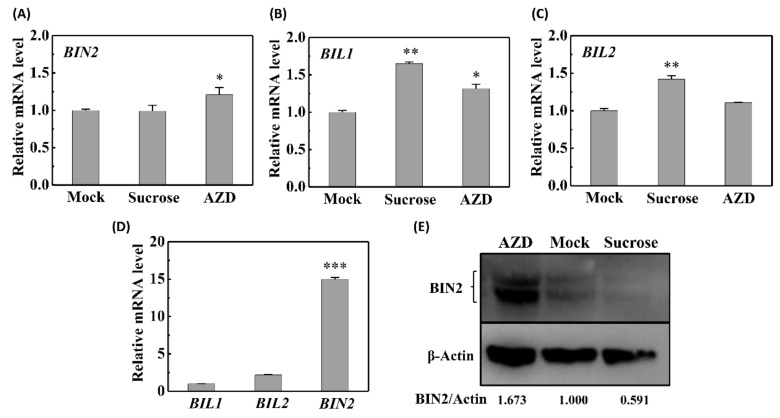
Sucrose influences the mRNA levels of *GSK3*s and the protein abundance of BIN2. Expression patterns of the *BIN2* (**A**), *BIL1* (**B**), and *BIL2* (**C**) genes in wild-type (WT) etiolated seedlings treated with 30 mM sucrose or 1 μM AZD. (**D**) Comparison of the mRNA levels of *BIN2*, *BIL1*, and *BIL2* in untreated WT etiolated seedlings. Values are presented as the means ± SD of three independent sets of experiments. (**E**) Immunoblot analysis of BIN2 protein in WT etiolated seedlings grown in the dark on medium supplemented with 30 mM sucrose or 1 μM AZD. Data were statistically evaluated using Student’s t-test, and the significance of differences (*, *p* < 0.05; **, *p* < 0.01; ***, *p* < 0.001) is indicated.

**Figure 5 ijms-22-13588-f005:**
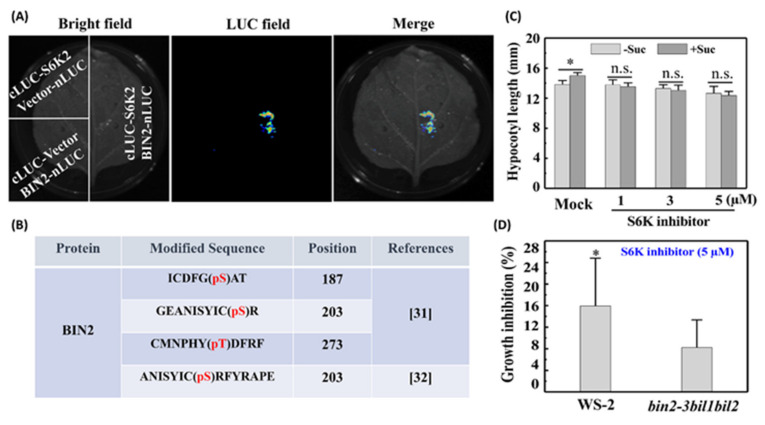
Roles of S6K2 in skotomorphogenesis. (**A**) Interaction between S6K2 and BIN2. The interaction was determined using a split-luciferase complementation assay in *Nicotiana benthamiana*. The full-length coding sequences of S6K2 and BIN2 were ligated into the pCAMBIA-35S-NLuc and pCAMBIA-35S-CLuc vectors, respectively. The empty vectors were used as negative controls. (**B**) In vitro phosphorylation sites of BIN2 catalyzed by S6K2 was identified by in vitro kinase assay followed by mass spectrometric analysis [31], of which S203 was found to be an in vivo phosphorylation site based on proteomics analyses [32] (http://epsd.biocuckoo.cn/ accessed on 21 May 2021). (**C**) Hypocotyl length of wild-type (WT) etiolated seedlings treated with different concentrations of S6K inhibitor (LY2584702 tosylate). (**D**) Growth inhibition of Ws-2 and *bin2-3bil1bil2* mutant induced by S6K inhibitor. Growth inhibition (%) = (1 − L_t_ / L_n_). L_t_ and L_n_ represent the hypocotyl length of etiolated seedlings grown in medium with or without the S6K inhibitor, respectively. Error bars represent the standard deviations (SD) (*n* > 20). Data were statistically evaluated using Student’s *t*-test, and the significance of differences (*, *p* < 0.05; n.s., not significant) is indicated.

**Figure 6 ijms-22-13588-f006:**
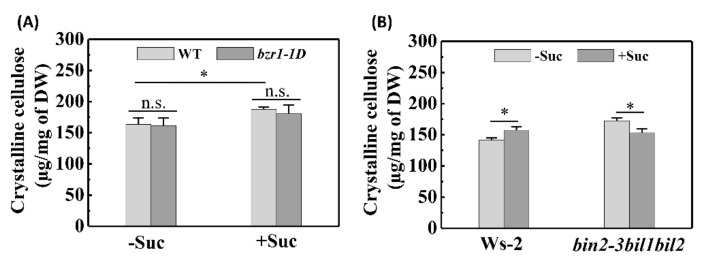
Measurements of crystalline cellulose content. (**A**) Crystalline cellulose contents of 5-day-old wild-type (WT) and *bzr1-1D* etiolated seedlings. (**B**) Crystalline cellulose contents of 5-day-old Ws-2 and *bin2-3bil1bil2* etiolated seedlings. Seedlings shown in both (**A**,**B**) were grown on medium with or without 30 mM sucrose. Values are presented as the means ± SD of three independent sets of experiments. Data were statistically evaluated using Student’s *t*-test, and the significance of differences (*, *p* < 0.05; n.s., not significant) is indicated.

**Figure 7 ijms-22-13588-f007:**
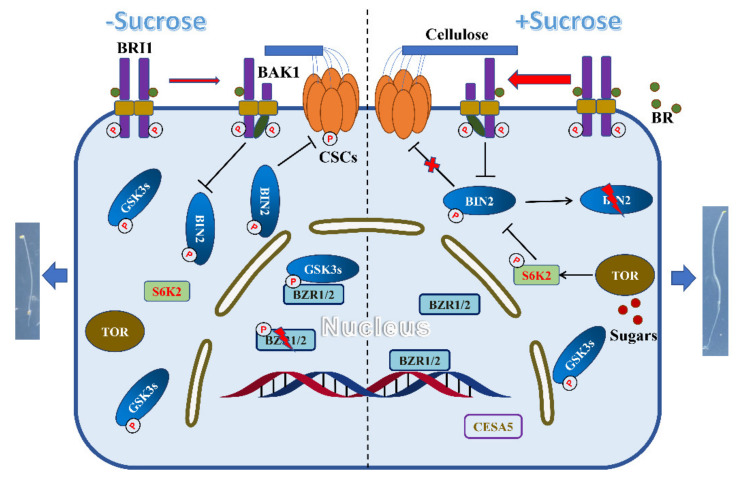
A model for sucrose-promoted skotomorphogenesis in the dark. The model illustrates the mechanisms whereby exogenously supplied sugar enhances brassinosteroid (BR) biosynthesis [29], the interaction between BRI1 and BAK1 [42], and the activation of the TOR-S6K2 pathway, resulting in the inhibition of GSK3 kinases and the enhancement of BIN2 degradation. Inactivation of BIN2 and its homologs leads to the accumulation of BZR1/2 for transcriptional regulation (including the upregulated expression of *CESA* genes [48]) and the enhancement of CESA complex activity [50] for cellulose synthesis, which may synergistically promote skotomorphogenesis in the dark.

## Data Availability

Data is contained within the article.

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
