# Peer review of "Exogenous Application of Low-Concentration Sugar Enhances Brassinosteroid Signaling for Skotomorphogenesis by Promoting BIN2 Degradation"

_ijms, 2021, doi:10.3390/ijms222413588_

Round 1
Reviewer 2 Report
The author studied cross-talk between sucrose and BR signaling with respect to hypocotyl elongation in the dark and established that BIN2 serves as a converging node that integrates the cross-talk between sugar and BR signaling in the regulation of plant growth. Accumulation of BZR1 in bzr1-1D mutant plants partially rescues the growth defects induced by the TOR inhibitor AZD, and that these plants display a normal phenotype similar to that of wild-type plants in response to both sucrose and non-sucrose treatments, thereby indicating that accumulated BZR1 functions, at least partially, in the sucrose-promoted elongation of hypocotyls. The manuscript is well structured and well discussed. However, some points should be checked and corrected. Therefore, I recommend major revision according to given my comments.
- The abstract is not clear. Please add the aim and objective of the review.
- Figures 4 and 5 please provide the statistical significance.
- Please speculate on the results. The discussion must improve.
- The MS English needs to be improved. The article's English must be carefully checked for grammatical errors.
- In Conclusion, the authors should add the significance of this review, and its potential practical application.
Round 2
Reviewer 1 Report
The authors have properly taken into account my critical concerns. Hopefully they stimulate the authors to go deeper into the complex cross-talks between sucrose, glucose and hormone signaling.
Reviewer 2 Report
The requested corrections were completed.